# Human Aligned Reward Modeling for Automated Transfer Function Generation of 3D Rendering of Medical Image Data

**David Melenberg**[*1]                    DAVID.MELENBERG@IMTE.FRAUNHOFER.DE
**Nele Blum**[*1]                              NELE.BLUM@IMTE.FRAUNHOFER.DE
**Cem Adiyaman**[1]                       CEM.ADIYAMAN@IMTE.FRAUNHOFER.DE
**Thorsten M. Buzug**[1,2]          THORSTEN.BUZUG@IMTE.FRAUNHOFER.DE
**Maik Stille**[1]                              MAIK.STILLE@IMTE.FRAUNHOFER.DE

[1] *Fraunhofer IMTE, Fraunhofer Research Institution for Individualized and Cell-Based Medical Engineering, Lübeck, Germany*

[2] *Institute for Medical Engineering, University of Lübeck, Lübeck, Germany*

**Editors:** Accepted for publication at MIDL 2025

## Abstract

We propose a reinforcement learning framework to automate the design of 2D transfer functions for direct volume rendering of medical images. By training a reward model based on human feedback, our approach enables an agent to extract transfer functions from joint histograms without manual fine-tuning. Preliminary results demonstrate that the developed method effectively captures human preferences, marking a significant step toward automated, user-aligned 3D renderings for improved patient communication, diagnosis, and treatment planning.

**Keywords:** Deep Learning, Human Feedback, Transfer Function, Volume Rendering

## 1. Introduction

Direct volume rendering (DVR) uses transfer functions (TFs) to map volumetric data to optical properties without prior surface extraction. Designing an effective TF is often unintuitive, repetitive, and time consuming (Pfister et al., 2001). It requires expertise to select features in 2D joint histograms (JHs) representing image characteristics. The manual TF design demands adaptation to new data. Iterative approaches are limited by their focus on opacity parameters and inability to incorporate neighboring information. Convolutional neural networks often require labeled data (Kim et al., 2021). To overcome these challenges, we propose a reinforcement learning from human feedback (RLHF) framework (Christiano et al., 2017). Instead of defining objective functions, RLHF utilizes human preferences to train an RL agent. In this work, we develop a reward model (RM) in an RLHF framework to automate the generation of 2D TF. We adopt the RLHF pipeline proposed by Ziegler et al. (2020), which includes three phases: the supervised fine-tuning of an agent, the preference collection for a subsequent RM training, and the RL fine-tuning using proximal policy optimization (PPO) (Schulman et al., 2017).

---

[*] Contributed equally

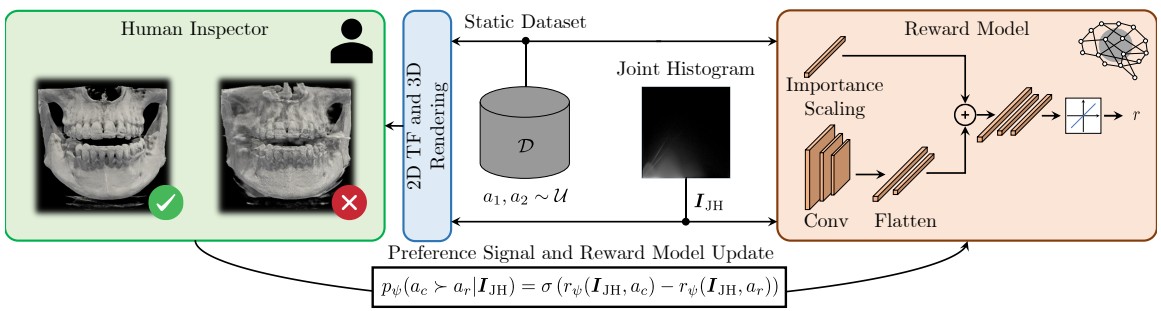

Figure 1: RM training concept in an RL framework for generating 2D TFs.

## 2. Methods

In our RL framework, the environment's state is defined as $s = \boldsymbol{I}_{\text{JH}}$, where $\boldsymbol{I}_{\text{JH}}$ is the image of the JH over the intensities and gradients of the original 2D slices of the input volume. The agent's action is to estimate suitable vertices of a polygon within this JH, used to compute the 2D TF for DVR. The action is defined as $a = [(x_i, y_i)]_{i=1,\ldots,n}^{\text{T}}$, where $x_i$ and $y_i$ are the coordinates of the polygon's vertices within the JH image with $n = 4$ to simplify the initial action space. The RL framework is implemented as a one-shot learning method allowing the agent to find the vertices of the polygon in a single time step per episode. The RM predicts the quality of an action given the state, guiding the agent towards actions that align the resulting 3D rendering with human visual preferences. Figure 1 illustrates the concept and architecture of the RM. For offline training, we exclude the agent and use a static dataset $\mathcal{D}$ containing predefined human-labeled and randomly generated actions. The RM receives the JH image and the action as input. The JH image is processed through convolutional layers, each with a dropout probability of $p = 50\,\%$ to extract image features, while the action features are upscaled to maintain balance. Both features are concatenated and further processed to predict the scalar reward $r$. The model is trained using human feedback on rendering pairs generated from different actions. These are compared by human inspectors expressing a preference, resulting in $a_c \succ a_r$, where $a_c$ and $a_r$ represent the chosen and rejected actions, respectively. Following the Bradley-Terry model (Bradley and Terry, 1952) for estimating score functions from pairwise preferences, the RM $r_\psi$ aims to satisfy

$$p_\psi(a_c \succ a_r | s) = \frac{\exp(r_\psi(s, a_c))}{\exp(r_\psi(s, a_c)) + \exp(r_\psi(s, a_r))} = \sigma(r_\psi(s, a_c) - r_\psi(s, a_r)), \qquad (1)$$

where $\sigma$ is the sigmoid function. We train the RM using the cross-entropy loss function

$$\mathcal{L}_C = -\mathbb{E}_{\mathcal{D}} \left[ \mu_c \log p_\psi(a_c \succ a_r) + \mu_r \log p_\psi(a_r \succ a_c) \right], \qquad (2)$$

where $\mu$ indicates the human inspector's preference distribution over $\{c, r\}$ introduced by Christiano et al. (2017). We utilized a proprietary dataset containing 16 CBCT head images focused on the cranial region, each with dimensions $547 \times 421 \times 547$ and a pixel spacing of $0.2\,\text{mm}$. Initial tests were performed on one image scene with 50 random and 10 predefined actions based on manual TF designs indicating high-quality renderings. An additional 40 actions were generated by introducing small random shifts to the polygon vertices, resulting

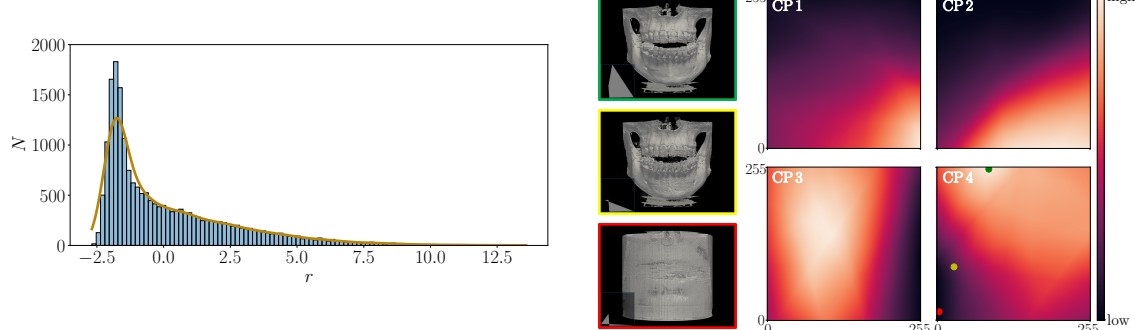

Figure 2: Calculated rewards by the trained RM. Corner points plotted for each vertex of the polygon from the calculated 2D TF with embeddings derived from CP 4.

in 100 actions. Human inspectors compared 4000 rendering pairs generated from these actions using a preference interface (3387 unambiguous and 613 ambiguous). The RM was implemented in PyTorch (Paszke et al., 2019) and trained for 75 epochs with a batch size of 128, using the Adam optimizer (Kingma and Ba, 2015) with the default learning rate.

## 3. Results and Discussion

Figure 2 shows the reward distribution (left) calculated for 20 000 uniformly distributed random actions on the test scene, which has a high frequency of low rewards, sharply declining toward higher rewards. This is the expected behavior of the RM since the majority of actions would result in poor visualizations and only actions leading to preferred renderings should increase the reward. Additionally a corner point plot (right) was created for each vertex of the polygon defining the TF, modifying one vertex at a time while keeping others fixed based on a high-quality rendering. They show the RM's calculated reward for each position in the JH. This plot indicates that only narrow ranges yield high rewards, showing effective discrimination between good and poor actions. Increasing the number of actions with smaller deviations in critical JH areas and collecting more human preferences could enhance the RM's precision. Expanding the dataset to include multiple scenes can improve generalization. Gathering preferences from multiple users would further enhance objectivity.

## 4. Conclusion

We have taken an important step toward automating 2D TF generation for optimized volume rendering by developing an RM trained on human feedback. The RM successfully captures human visual preferences, validating its suitability for guiding an RL agent. Future work will focus on training the RM on more data to develop the full RLHF pipeline.

## Acknowledgments

Funded by Land Schleswig-Holstein through the Project "Individualisierte Medizintechnik für bildgestützte, robotische Interventionen (IMTE 2)", Project number: 125 24 009.

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

## Appendix A. Additional Evaluation Visualization

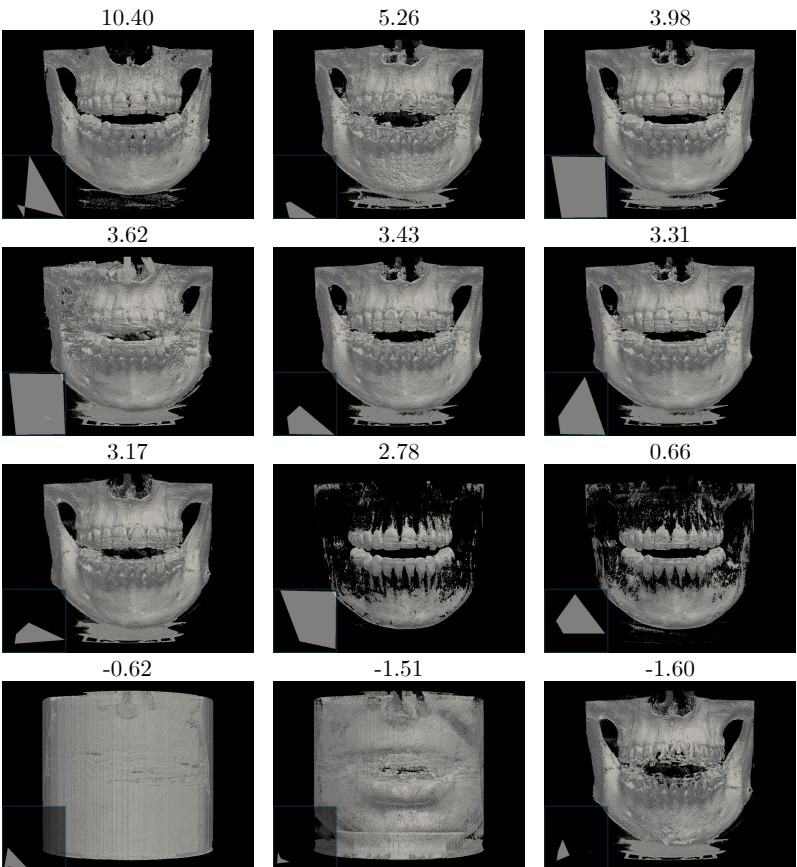

Figure 3: Example results from the RM with scores from highest to lowest.

Figure 3 shows the predicted rewards based on 12 additional labeled evaluation TFs. Poor rendering results receive a low reward, while the reward is typically higher for renderings with a clearer visual representation of the jaw. Despite the generally good assignments of the rewards, there are still individual outliers. For example, the rendering in the second row and first column still receives a comparatively high reward, despite some artifacts in the dental area.

