# OpenReview forum: "Human Aligned Reward Modeling for Automated Transfer Function Generation of 3D Rendering of Medical Image Data"
_MIDL.io/2025/Short_Papers — MIDL 2025 - Short Papers_

### Official Review · Reviewer_pxXc · 2025-04-16

**Rating:** 4
**Confidence:** 4

**Summary:**

This paper proposes a reinforcement learning from human feedback framework to automate 2D transfer function design for direct volume rendering of medical image data. The authors develop a reward model trained on human preference data, guiding an agent to generate TFs by selecting polygon regions in joint histograms. Their approach eliminates the need for manual TF tuning and aims to produce renderings that better align with human visual preferences. Preliminary results on a proprietary CBCT dataset show the RM can effectively differentiate between high- and low-quality renderings.

**Strengths:**

- The integration of preference-based reward modeling into the transfer function design problem is both original and well-motivated.
- Manual TF tuning for DVR is known to be complex and subjective. This framework addresses that bottleneck effectively.
- Emphasizing alignment with visual preferences makes the approach particularly relevant for communication and interpretation tasks in clinical practice.
- The RM is trained using the Bradley-Terry model and validated on a significant number of rendering comparisons (4000), showing a good solid implementation.

**Weaknesses:**

- The current study is confined to one CBCT image scene for training/testing and relies on a relatively small set of TF actions.
- Only the RM is evaluated; the full RLHF pipeline (with PPO fine-tuning of the agent) is planned but not yet implemented.
- While human alignment is a strength, preference labels are inherently subjective and potentially inconsistent without multiple raters.
- Results are mainly visual and reward-distribution-based. more quantitative or perceptual quality metrics could strengthen the validation.

---

### Decision · Program_Chairs · 2025-05-01

Accept